# Knowledge, Awareness, Attitudes and Practices toward Perimenopausal Symptoms among Saudi Females

**DOI:** 10.3390/healthcare12060677

**Published:** 2024-03-18

**Authors:** Mohammed A. Aljunaid, Lojain Nasser Alruwaili, Hamzah Yahya Alhajuj, Mohammed Talal Musslem, Hussain Hasan Jamal

**Affiliations:** 1Department of Family and Community Medicine, University of Jeddah, Jeddah 21589, Saudi Arabia; 2Faculty of Medicine, University of Jeddah, Jeddah 21589, Saudi Arabia; lojain8314@gmail.com (L.N.A.);; 3King Abdullah Medical Complex, Ministry of Health (Moh), Jeddah 23816, Saudi Arabia; hjammal@moh.gov.sa

**Keywords:** perimenopause, knowledge, awareness, practice, attitudes, Saudi

## Abstract

Women’s knowledge about perimenopause plays a crucial role in shaping their perception of related illnesses, influencing coping strategies, treatment adherence, and the overall management of this life stage. This cross-sectional study assessed the awareness, knowledge, attitudes, and practices regarding perimenopause among 409 Saudi women attending primary healthcare centers. Participants completed a structured questionnaire addressing demographic data, awareness, knowledge, attitudes, and practices related to perimenopause. While 75.3% of the participants were aware of perimenopause, only 17.4% could identify more than 10 out of 20 perimenopause symptoms. Commonly recognized symptoms included menstrual irregularity (67.7%), mood swings (66.0%), and mood fluctuations (50.4%). Only 23.0% had optimal knowledge about perimenopause complications. Additionally, 73.3% had not consulted a doctor for perimenopause-related issues. An analysis of the overall knowledge score showed a mean (SD) = 14.82 (5.64) out of 34. The level of knowledge was independently associated with a higher educational level, more frequent perimenopause symptoms, and regular doctor visits. This study reveals high awareness but insufficient knowledge among Saudi women regarding perimenopause symptoms and complications associated with higher perimenopause morbidity and a lack of engagement with healthcare professionals. It underscores the need for early and continued education on perimenopause, improved doctor–patient communication, and specific interventions to boost knowledge and attitudes toward perimenopause.

## 1. Introduction

Perimenopause refers to the period of time surrounding the final years of a woman’s reproductive life. It extends from the first onset of menstrual irregularity to 12 months of menstrual interruption, which marks the final menstrual period (FMP) [1]. This period is marked by several hormonal and physiological changes, often accompanied by a wide range of clinical and mental symptomatology [2]. According to the Stages of Reproductive Ageing Workshop +10 (STRAW + 10), perimenopause (aka., menopausal transition) is subdivided into the following four phases: the early postmenopause (+1a, +1b, +1c), the early and late menopausal transition stages (−2 and −1) and late reproductive stages (−3b and −3a) [3]. Perimenopause starts at a median age of 47, progresses to late menopausal transition at a median age of 49, and arrives at the FMP at an age of 51–52, with a total duration of four or more years [1].

The clinical manifestations of perimenopause typically include vasomotor symptoms (hot flushes, palpitations, night sweats, and sleep disturbance), genitourinary syndrome (regrouping vaginal dryness/itching, sexual complaints, and urinary symptoms), psychological symptoms (depression, anxiety, irritability) as well as bleeding (i.e., menstrual) anomalies. Other manifestations, such as neuromuscular symptoms, are also common [1,2,4]. Hence, about 75% and 77% of women in the USA reported experiencing hot flashes and joint pain, respectively, during perimenopause [5,6]. A cross-sectional study performed in Saudi Arabia reported a higher frequency of symptoms in perimenopausal women compared to premenopausal and postmenopausal women [7]. Other data from Saudi Arabia demonstrated that perimenopausal women experience more severe symptoms than other groups [8]. Notably, perimenopause can result in deep repercussions on quality of life and daily activity. Additionally, the incidence of multiple morbidities such as cardiovascular disease, breast cancer, osteoarthritis increases considerably during the period of perimenopause compared to postmenopause [9]. 

In recent years, perimenopause and menopause have gained more attention due to an increase in life expectancy and population growth. The number of menopausal and postmenopausal women is expected to reach 1.2 billion by 2030 [10]. Moreover, women are expected to live approximately more than one-third of their lives as menopausal [11]. Therefore, effective strategies are needed to reduce morbidity and improve the overall quality of life among perimenopausal women. 

In line with this, women’s level of knowledge is a fundamental contributor to the enhancement of subjective illness perception, coping strategies, treatment compliance, and adequate outcomes of disease management. Particularly, an awareness level about menopausal symptoms and ways of coping with them were found to be critical for women to access healthcare during the menopausal period [12]. Women with more negative attitudes towards menopause in general reported more symptoms during the menopausal transition [13]. In Saudi Arabia, there is a lack of data regarding whether women are sufficiently informed about perimenopause and the factors that determine their knowledge on this topic. Therefore, we conducted the present cross-sectional study to explore the knowledge and attitudes of perimenopausal Saudi females. Such data can help identify any existing gaps in women’s education and behavior, which could ultimately help improve perimenopause management and decrease its impact on quality of life.

## 2. Materials and Methods

### 2.1. Design and Setting

This was a cross-sectional study that was conducted at the Ministry of Health (MOH) primary care centers (PHCs) of Jeddah, Saudi Arabia, between 3 September and 26 October 2023. 

### 2.2. Participants

The study involved Saudi women aged 35–50 years who attended the participating PHC clinics during the study period. The study excluded women who refused to be interviewed, were healthcare workers, or had any of the following conditions: menopause, psychiatric issues, gynecological diseases, or were outside the target age range.

### 2.3. Sample Size

The sample size was calculated using the Raosoft sample size calculator. Based on an error margin of 5%, a confidence interval of 95%, and an unknown awareness rate (presumed 50%), the initial sample size was determined to be 385 patients. To account for non-respondents and incomplete participation, an additional 10% (38 patients) was added, bringing the total estimated sample size to 423 patients. A convenience sampling method was used to include all consecutive consenting and eligible women until reaching the target sample size.

### 2.4. Sampling

In Jeddah city, there are 49 PHC centers (PHCs). A random number generator (http://www.random.org, accessed on 28 August 2023) was used to select 30 out of the 49 PHCs based on a list obtained from the Public Health Administration. We estimated the average number of patients per PHC by dividing the total number of patients (423) by the total number of PHCs (30), which resulted in approximately 14–15 patients per participating PHC.

### 2.5. Data Collection Tool

A structured, self-administered questionnaire was designed by the authors to collect the following data: 

1. Demographic and health-related data, including basic personal information such as age, nationality, educational level, etc., as well as general and reproductive health-related data such as comorbidities, contraceptive use, smoking status, etc.

2. Awareness about perimenopause: this section explored the participants’ awareness of perimenopause, their sources of information, and whether they had actively sought information about it. 

3. Knowledge about perimenopause: this section explored in-depth knowledge about perimenopause, focusing on the following 3 dimensions: (a) basic knowledge exploring the participants’ understanding of perimenopause, its causes, and impact (4 items); (b) symptoms including a list of 20 items such as irregular menstruation, mood swings, etc.; and (c) associated health risks and complications including a list of 10 items such as obesity, daily life activity impact, etc.

4. Attitudes and practices in perimenopause: This final section evaluated the participants’ emotional response to perimenopause (e.g., excited, resigned, indifferent), whether they sought medical advice for it, the frequency of their follow-up visits regarding perimenopause, and the variety of perimenopause symptoms experienced. Furthermore, this section explored the methods used to mitigate symptoms (cool showers, use of vaginal moisturizers, losing weight, etc.), beliefs about the effectiveness of lifestyle changes, and opinions on the best timing for perimenopause education. 

The questionnaire underwent face and content validity by two family physicians. The final version comprised a total of 45 questions.

### 2.6. Procedure

Data collection for the study involved the use of both hard-copy questionnaires and online self-administered surveys via Google Sheets, which were accessed on digital tablets supplied by the researchers. The research team visited participating PHCs to identify eligible participants, who were approached upon completion of their physician’s visit and informed about the study. Following a detailed explanation of the study’s objectives and obtaining verbal consent, participants were given the choice between an online (smart tablet) or a hard copy of the questionnaire, based on their preference. Nonetheless, all participants opted for the digital version for its convenience. Data collection took place in a dedicated room within the clinic, supervised by a research team member who was available to clarify any ambiguous questions. A 20 min timeframe was set for completion, after which the questionnaires were collected on Google Drive. This process ensured structured and efficient data collection. The team continued in each center until reaching the target of 14–15 participants per center.

### 2.7. Ethical Clearance

The data collection procedure ensured compliance with ethical standards, including respect for autonomy, privacy, and nonmaleficence. Informed verbal consent was sought from each participant prior to participation. Additionally, participants were informed of their right to withdraw from the study; however, because data collection was anonymous, withdrawal was only feasible before questionnaire submission. The study protocol and tool were reviewed and ethically approved by the Bioethics Committee of Scientific and Medical Research, Directorate of Health Affairs, MOH, Jeddah (#A01578). 

### 2.8. Statistical Analysis

Statistical analysis was performed with the Statistical Package for Social Sciences version 21.0 for Windows (SPSS Inc., Chicago, IL, USA). Categorical variables were presented as the frequency and percentage, while continuous variables were presented as the mean ± standard deviation (SD). 

The level of knowledge was assessed using the number of correct responses across the following three subscales: basic knowledge (4 items), symptoms (20 items), and complications (10 items). The construct validity of this scale was analyzed using Principal Component Analysis (PCA) followed by Direct Oblimin Rotation. The suitability of the dataset for factor analysis was assessed using the Kaiser–Meyer–Olkin (KMO) measure of sampling adequacy (value = 0.776) and Bartlett’s test of sphericity (statistics = 2686.11; *p* < 0.001), indicating suitability for factor analysis. The initial extraction, considering an initial eigenvalue ≥ 1 and an extraction value above 0.5, showed 11 extracted components, explaining a total variance of 57.7%. The first component showed an eigenvalue = 4.92 and explained 14.46% of the variance, while the second showed an eigenvalue = 2.63, explaining an additional 7.74% of the variance. The analysis of the Scree plot, as well as the comparison of calculated eigenvalues with those of the Monte Carlo PCA for parallel analysis, suggested that this scale has a multidimensional construct. Nonetheless, analysis of the component matrix showed higher loading for all but one item (perimenopause is caused by infertility) on the first component, indicating a strong association with a unidimensional construct. 

Additionally, the overall knowledge scale demonstrated reliability, as evidenced by Cronbach’s alpha of 0.806 for the 34 items. This indicates that the scale has an acceptable one-dimensionality and is reliable for scoring. Given that each item is dichotomous (yes or no), a single point was assigned for each correct response. Consequently, the overall knowledge score, ranging from 0 to 34, was computed as the total of these individual scores. The normal distribution of the overall knowledge score was assessed using the one-sample Kolmogorov–Smirnov Z test. 

Factors associated with knowledge were analyzed by comparing the overall knowledge score variance across different factors’ categories using the independent *t*-test or one-way ANOVA test, as applicable. Linear regression was also used to analyze the association between the knowledge score and age at the first menarche. A stepwise multivariate linear regression model was carried out to analyze the independent factors associated with the knowledge score; results are presented as regression coefficient B with a 95% confidence interval (95%CI).

A *p*-value of <0.05 was considered to reject the null hypothesis.

## 3. Results

### 3.1. Participant’s Demographic and Health-Related Characteristics

The study involved 409 Saudi females. The mean age was 42.20 years (SD = 4.67), and the mean age at first menarche was 12.88 years (SD = 1.73). The majority were married (72.9%), while singles, divorced, and widowed participants constituted 8.1%, 15.4%, and 3.7%, respectively. Education levels were relatively high, with 48.4% having university/bachelor’s degrees and 7.1% as postgraduates. Most participants were housewives (55.5%), and 40.3% were employed. Regarding income, 34.0% reported none, and the rest varied across income brackets. The majority had 1–4 children (59.9%), and 72.9% were non-smokers. Contraceptive use was reported by 45.7%, with hormonal methods being the most common (32.5%). Comorbidities were present in 48.7%, with diabetes (18.3%), hypertension (17.4%), and hyperlipidemia (11.7%) being the most common. Around half (51.3%) had no comorbidities. Medication adherence varied, with 29.6% always adhering to their prescriptions (Table 1).

### 3.2. Awareness and Knowledge about Perimenopause

Awareness and knowledge about perimenopause are depicted in Table 2, while knowledge about specific symptoms and complications related to perimenopause is depicted in Figure 1 and Figure 2, respectively. 

The majority (75.3%) of the participants were aware of perimenopause, with social media (46.0%) being the primary source of information, followed by family (34.5%) and friends (27.4%). Despite this awareness, 68.5% had not actively sought information about perimenopause. 

Regarding basic knowledge, a high percentage acknowledged that permanent menstrual cycle interruption could be a natural aging phenomenon (94.6%) and that physiological changes during perimenopause impact a woman’s body (84.6%), while two-thirds (68.9%) acknowledged that menstrual cycle interruption could be caused by a disease. On the other hand, only 36.7% associated menstrual cycle interruption with infertility.

However, only 17.4% demonstrated the optimal identification of more than 10 perimenopause symptoms out of the 20 listed. The most commonly identified symptoms were menstrual cycle irregularity (67.7%), mood swings (66.0%), and bad mood (50.4%), whereas fractures (10.5%), urinary tract symptoms (12.5%), and dryness and itchiness (19.6%) were the least commonly identified. 

Regarding complications, 34.5% identified fewer than four out of the ten listed symptoms, while 23.0% recognized eight or more. Stroke, cardiopathy, and urinary incontinence were the least frequently identified complications, acknowledged by 19.8%, 35.7%, and 50.1% of the participants, respectively.

### 3.3. Attitudes and Practice in Perimenopause

Attitudes towards perimenopause varied, with 36.9% of participants feeling indifferent, 27.4% resigned, and 24.2% worried. Regarding practices, the majority (73.3%) had not visited a doctor for perimenopause-related issues. Regular follow-ups were infrequent, with 45.7% never following up and only 6.6% reporting regular follow-up. Regarding experience, 18.3% reported experiencing no perimenopause symptoms, while 63.4% reported three or more symptoms (Table 3). The most frequently reported symptoms were mood swings (57.0%), bad mood (37.7%), and weight changes (36.9%) (Figure 3). In terms of symptom management, exercising (49.1%) and losing weight (34.0%) were the most common strategies, followed by using cool showers and cold drinks (27.6%). As for the effectiveness of lifestyle changes, 39.4% believed them to be effective, while 32.3% were unsure. Most participants (50.1%) thought perimenopause education should be included in medical routine checks, and 46.2% advocated for its inclusion at the university level (Table 3).

### 3.4. Knowledge Level about Perimenopause

An analysis of the knowledge score distribution using the one-sample Kolmogorov–Smirnov Z test showed statistics = 1.11 (*p* = 0.169), indicating that the ‘Overall Knowledge Score’ variable did not significantly deviate from normality. The analysis of the overall knowledge score showed a mean (SD) = 14.82 (5.64) out of 34 and a median (IQR) of 15 (7).

### 3.5. Demographic and Health-Related Factors Associated with Knowledge Score

Neither age nor age at first menarche was significantly associated with knowledge scores (*p* = 0.223 and *p* = 0.754, respectively). Among marital statuses, divorced participants had the highest mean knowledge score (15.51), but these differences were not statistically significant (*p* = 0.151). Educational level showed a significant association with knowledge scores; postgraduates had the highest mean score (16.00), followed by those with a university/bachelor’s degree (15.87), with significance at *p*-value < 0.001. Monthly income also displayed a significant correlation; participants with monthly incomes of SAR 5–10 K had the highest mean knowledge score (16.37), and this association was significant (*p* = 0.014). Parity, smoking status, and contraceptive use did not show significant associations with knowledge scores (*p* = 0.207, *p* = 0.104, and *p* = 0.396, respectively). Participants with two or more comorbidities scored higher on average (16.40), with this difference reaching statistical significance (*p* = 0.031). No significant associations were observed between knowledge scores and profession, prescribed medication, or medication adherence (Table 4).

### 3.6. Association of Knowledge about Perimenopause with Attitudes and Practice

Feelings about perimenopause were significantly associated with knowledge scores, with the highest mean score observed in those who were worried (15.71) or undecided (15.28), though no significant association was found (*p* = 0.400). Participants who visited a doctor for perimenopause-related issues had a significantly higher knowledge score (16.27 versus 14.29) compared to those who did not, respectively (*p* = 0.002). The frequency of follow-up visits also correlated significantly with knowledge; participants who followed up regularly had the highest knowledge score (18.11), with a statistically significant association (*p* < 0.001). The number of symptoms experienced showed a positive correlation with knowledge; those experiencing more than five symptoms had the highest mean knowledge score (17.35), which was statistically significant (*p* < 0.001). Additionally, the number of lifestyle measures practiced was positively associated with knowledge, with those practicing three or more measures having a significantly higher knowledge score (16.77), whereas those practicing no lifestyle measures had the lowest score (11.52); the difference was statistically significant (*p* < 0.001). A belief in the effectiveness of lifestyle changes in perimenopause also correlated significantly with knowledge, with those who believed in their effectiveness having higher knowledge scores (*p* < 0.001) (Table 5).

### 3.7. Independent Factors Associated with Knowledge about Perimenopause

The stepwise multivariate linear regression showed that the knowledge score was independently associated with women’s educational level (B= 0.76; 95%CI: 0.39–1.56; *p* = 0.001), the number of symptoms experienced (B = 0.68; 95%CI: 0.19–1.67; *p* = 0.006), and the frequency of perimenopause symptoms during follow-up (B = 1.82; 95%CI: 1.28–2.37; *p* < 0.001). This model explained 15.4% of the knowledge score variance, indicating the significant role of these factors in predicting knowledge. 

## 4. Discussion

### 4.1. Summary of Findings

Understanding how Saudi women perceive and manage perimenopause is crucial for developing tailored health interventions and educational programs. This study investigated the awareness and knowledge of perimenopause, as well as the associated attitudes and practices among Saudi women. Given the cultural and societal nuances influencing health perceptions and behaviors in Saudi Arabia, this study offers valuable insights into how these factors shape women’s understanding and responses to perimenopause. Such data further help identify gaps and opportunities for enhancing women’s health education and support systems locally and in their region.

Data from 409 Saudi females attending PHC clinics revealed substantial awareness (75.3%) about perimenopause, primarily through social media and relatives. Despite this level of awareness, a significant proportion (68.5%) had not actively sought information on the subject. Participants demonstrated varying degrees of knowledge, with the highest awareness observed in areas like menstrual cycle interruption as a natural aging phenomenon (94.6%). However, there was a notable gap in recognizing perimenopause symptoms and complications. Attitudes towards perimenopause varied, with indifference being the most common (36.9%). Practices to manage perimenopause were diverse, but only a minority (26.7%) visited a doctor for perimenopause-related issues. Among the most significant factors, higher knowledge was independently associated with a higher education level, experiencing more frequent perimenopausal symptoms, and more regular follow-ups for these symptoms. This study underscores the need for targeted educational efforts to bridge knowledge gaps and promote proactive health-seeking behaviors in perimenopause management.

### 4.2. Role of Social Media/Internet in Perimenopause Knowledge Acquisition

The study participants showed high perimenopause awareness despite limited active information seeking. In fact, our findings reaffirm the role played by social media as a popular tool to seek medical information for different topics, including perimenopause. For a sensitive topic such as perimenopause, women may prefer to self-educate through online platforms or discussions with friends, as this may provide a more private information-seeking context. Notably, in a survey including data from 947 perimenopausal women, Harper et al. found that 68.2% of the interviewed participants had looked for information about menopause mainly by talking to friends and using a variety of websites [14]. Another study carried out by Tariq et al. identified websites and friends as the most popular sources of menopause information among 3149 women [15]. Saudi women prefer the internet and social media to gather medical information much more than visiting a doctor [16]. This suggests the necessity to optimize the scientific reliability of telehealth resources, particularly social media.

### 4.3. Inadequate Information Seeking about Perimenopause

On the other hand, our study highlights fewer tendencies in Saudi women to display information-seeking behavior from healthcare professionals (HCPs) in regard to perimenopause. In contrast, women from other countries, such as the UK, were found to have high rates of menopause symptoms disclosure to HCPs [17]. In line with this, interviewed students from the UK described peri/menopause as “a medical problem to be easily and effectively resolved by a doctor” [18]. Therefore, there is a need to explore factors that determine active perimenopause knowledge in Middle Eastern societies. Reported barriers to seeking help for menopausal symptoms include a lack of knowledge of the symptom’s full range, stigma, embarrassment, and normalization of the menopause phenomenon [19].

### 4.4. Gaps in Perimenopause Knowledge among Saudi Women

The lack of adequate knowledge regarding perimenopause leaves women passively experiencing this sensitive life stage without anticipated coping strategies as well as health-related measures that could reduce the physical and psychological consequences of reproductive and menstrual activity loss. The vast majority of study participants appear to understand that perimenopause is a natural phenomenon in their lives at which they lose the ability to conceive, and this may be accompanied by corporeal modifications. In contrast, it was previously reported that more than 84% of women believe that perimenopausal symptoms are related to health disturbances [20]. Nonetheless, the potential pathological aspect of perimenopause was under-recognized by the Saudi women in our study, which probably favored their lower reactivity to symptoms, with only a minority reported visiting a doctor. Thus, we identified existing gaps in the knowledge about perimenopause-related manifestations and complications, suggesting that the quality of information delivered about this phenomenon and its consequences is still insufficient.

### 4.5. Impact of Low Knowledge on Attitudes towards Perimenopause

The present study showed knowledge levels to be independently associated with both the number of symptoms experienced and the frequency of doctor visits for these symptoms. This suggests that women’s knowledge mainly increases following high morbidity that necessitates frequent visits to the doctor. On the other hand, suboptimal levels of knowledge are likely to result in higher perimenopause-related morbidity due to a lower understanding of what symptoms should be interpreted as relatively normal (e.g., hot flushes and vaginal dryness) and those that should drive more inquietude (e.g., pelvic pain). Moreover, misinformed women are less likely to be visiting doctors during perimenopause, which may promote poorer management of this period’s associated syndromes. This underscores the importance of systematic health education on perimenopause, irrespective of the symptom level experienced, to enhance awareness and foster better care-seeking behavior. Indeed, perimenopausal women reporting poor quality of care showed greater symptom misinformation and misattribution [17]. Women not aware of the impact that their lifestyle can have on their health during perimenopause may reduce their physical activity (27.4% of the participants resigned in our study) while continuing to adopt unhealthy habits, which can increase cardiovascular risks at this stage of life. Furthermore, the level of perimenopause knowledge may influence the proper use of hormonal therapy. It has been shown that the greater the knowledge of menopause, the better the knowledge of hormonal therapy [21].

### 4.6. Association between Educational Level and Perimenopause Awareness

We have also reaffirmed the significant association between education and the level of perimenopause knowledge, showing an independent relationship in the stepwise linear regression model. Thus, it has been previously found that a higher education level was associated with greater perimenopause-related knowledge in midlife women [20]. Education has not only an influence on awareness but also attitudes. Hence, attitudes toward perimenopause are generally negative and most likely are worsened in the absence of education and literacy on health [22].

### 4.7. Implications for Public Health and Education

There is a need for educational interventions to enhance positive attitudes toward the perimenopause period in Eastern societies. School can be a suitable place to provide the necessary information at an early age as women often receive no education about peri/menopause in the school setting [14,15]. Involvement in educational programs can improve women’s attitudes toward perimenopause by enhancing their self-management abilities. Hence, a structured educational intervention was previously shown to be beneficial in intensifying women’s knowledge and positively impacting their attitudes during perimenopause [23]. Therefore, implementing more perimenopause education strategies is warranted in Saudi females, and they should start receiving adequate knowledge at school and continue to receive it beyond school. 

Additionally, strategies are required to enhance women’s tendency to disclose their perimenopause experience to doctors and seek information about the management of symptoms. Women, especially in conservative societies, should be encouraged to undergo a “perimenopause visit” in which they receive the proper care and education regarding perimenopause and hormonal therapy. Disclosing perimenopause symptoms can be embarrassing for women; therefore, doctors should show empathy and a sense of communication. It would also be an opportunity to evaluate the cardiovascular risk factors in perimenopausal women and recommend any required screenings (e.g., mammography for women at risk of breast cancer, hypertension, osteoporosis, etc.).

### 4.8. Study Limitations

The main limitation of this study is the cross-sectional design, which hinders establishing a causal relationship between knowledge and practice. Additionally, the study participants had a median age of over 40, which might suggest a higher level of knowledge related to perimenopause, as women under 40 tend to exhibit a more pronounced lack of information on this topic [15,21,24]. Finally, the definitive construct validity of the knowledge questionnaire was not confirmed, potentially affecting the internal validity of the findings. Further refinement of the knowledge questionnaire is necessary to improve its construct validity.

## 5. Conclusions

While awareness of perimenopause among Saudi women is high, many Saudi women lack adequate knowledge about symptoms and complications. This is associated with greater perimenopause-related morbidity and limited active information seeking, notably from healthcare professionals. This study also reveals a correlation between educational level and perimenopause knowledge, emphasizing the importance of integrating perimenopause education into early schooling and continuous health literacy programs. These findings highlight the importance of enhancing doctor–patient communication to encourage women to seek professional advice and manage perimenopause effectively. Furthermore, they underscore the necessity of targeted interventions to improve knowledge and attitudes towards perimenopause, fostering better health outcomes for women in this life stage.

## Figures and Tables

**Figure 1 healthcare-12-00677-f001:**
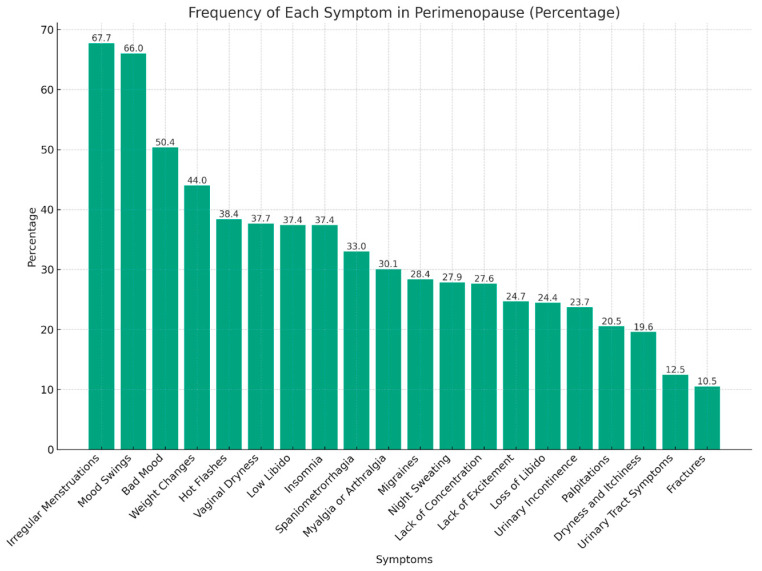
Knowledge about perimenopause symptoms (N = 409). Caption: Bars represent the percentage of participants who identified the given symptom as a sign of perimenopause.

**Figure 2 healthcare-12-00677-f002:**
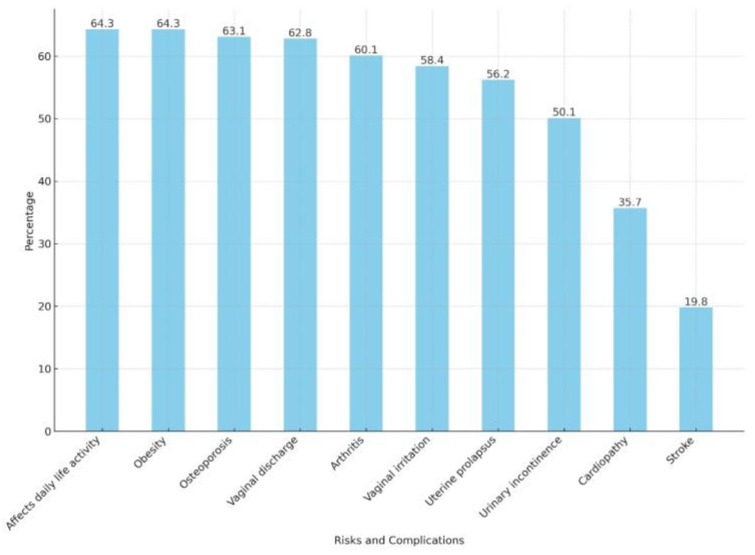
Knowledge about perimenopause risks and complications. Caption: Bars represent the percentage of participants who identified the given risk factor or complication as related to perimenopause.

**Figure 3 healthcare-12-00677-f003:**
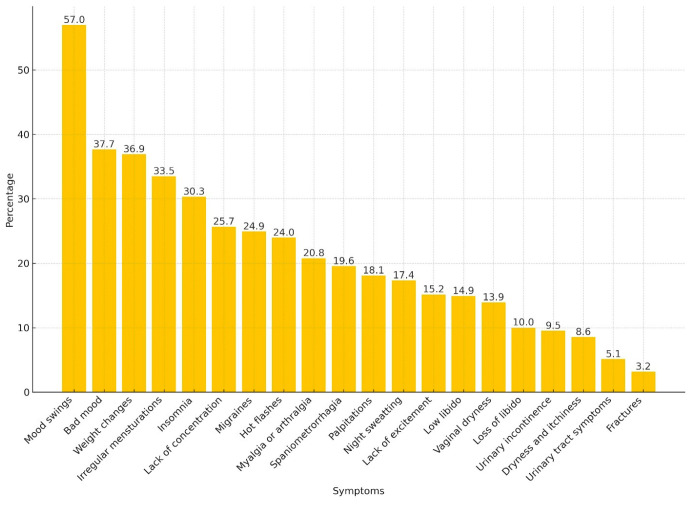
Experienced symptoms of perimenopause. Caption: Bars represent the percentage of participants who declared having experienced the given symptom of perimenopause.

**Table 1 healthcare-12-00677-t001:** Participant’s demographic and health-related characteristics (N = 409).

**Parameter**	**Level**	**Mean**	**SD**
Age	Years	42.20	4.67
Age at first menarche	Years	12.88	1.73
**Parameter**	**Level**	**Frequency**	**Percentage**
Marital status	Single	33	8.1
	Married	298	72.9
	Divorced	63	15.4
	Widowed	15	3.7
Education	Illiterate/Primary	12	2.9
	Middle school	35	8.6
	Secondary school	135	33.0
	University/Bachelor	198	48.4
	Postgraduate	29	7.1
Profession	Housewife	227	55.5
	Employed	165	40.3
	Retired or unemployed	17	4.2
Monthly income	No monthly income	139	34.0
	<1 K SAR	36	8.8
	1–5 K SAR	71	17.4
	5–10 K SAR	73	17.8
	10–15 K SAR	64	15.6
	>15 K SAR	26	6.4
Parity	None	55	13.4
	1–4	245	59.9
	5 or more	109	26.7
Smoking status	Smoker	23	5.6
	Ex-smoker	17	4.2
	Passive smoker	71	17.4
	Non-smoker	298	72.9
Contraceptive use	No	222	54.3
	Yes	187	45.7
	Hormonal	133	32.5
	Condoms	26	6.4
	IUD	24	5.9
Comorbidities §	Diabetes	75	18.3
	Hypertension	71	17.4
	Hyperlipidemia	48	11.7
	Asthma	30	7.3
	Hyperthyroidism	8	2.0
	Hypothyroidism	39	9.5
	Cardiopathy	4	1.0
	Kidney disease	6	1.5
	Arthritis	15	3.7
	Other	4	1.0
No. of comorbidities	0	210	51.3
	1	129	31.5
	2+	70	17.1
Prescribed medication	No	217	53.1
	Yes	192	46.9
Adherence	As prescribed	121	29.6
	Sometimes skipping	59	14.4
	Often skipping	9	2.2
	Rarely as prescribed	3	0.7

§ A participant could provide more than one answer. IUD: Intra uterine device; SAR: Saudi Riyal.

**Table 2 healthcare-12-00677-t002:** Awareness and knowledge about perimenopause.

Parameter	Level	Frequency	Percentage
Awareness			
Awareness about perimenopause	Yes	308	75.3
	No	101	24.7
Sources of awareness §	Friend	112	27.4
	Family	141	34.5
	Social media	188	46.0
	Healthcare providers	70	17.1
	Curriculum	54	13.2
	Scientific research	36	8.8
	Podcasts	17	4.2
	Magazines and newspapers	15	3.7
	TV shows	35	8.6
	Others	4	1.0
Active search for information about perimenopause	As soon as symptoms started	43	10.5
	Long after symptoms started	24	5.9
	Before symptoms started	62	15.2
	Have not searched	280	68.5
Knowledge			
Basic knowledge	Menstrual cycle interruption can be a natural phenomenon caused by aging	387	94.6
	Menstrual cycle interruption may be due to infertility	150	36.7
	Menstrual cycle interruption may be caused by a disease	282	68.9
	Physiological changes that occur during perimenopause impact the woman’s body	346	84.6
Symptoms (No. symptoms identified out of 20) ^⁂^	Poor (<5)	145	35.5
	Suboptimal (5–10)	193	47.2
	Optimal (>10)	71	17.4
Complications (No. complications identified) ^⁂^	Poor (<4)	141	34.5
	Suboptimal (5–7)	174	42.5
	Optimal (8+)	94	23.0

§ A participant could provide more than one answer. ^⁂^ Detailed symptoms and complications identified are depicted in Figure 1 and Figure 2, respectively.

**Table 3 healthcare-12-00677-t003:** Attitudes and practice in perimenopause.

Parameter	Level	Frequency	Percentage
Attitudes towards perimenopause	Excited	22	5.4
	Resigned	112	27.4
	Indifferent	151	36.9
	Worried	99	24.2
	Undecided	25	6.1
Visited doctor for perimenopause	Yes	109	26.7
	No	300	73.3
Frequency of follow-up	Regularly	27	6.6
	Often	92	22.5
	Rarely	103	25.2
	Never	187	45.7
Symptoms experienced ^⁂^	None	75	18.3
	1–2	75	18.3
	3–5	132	32.3
	>5	127	31.1
Methods used to mitigate symptoms §	Wearing light clothes	126	30.8
	Avoiding hot places	99	24.2
	Cool showers and cold drinks	113	27.6
	Vaginal moisturizers or lubricants	39	9.5
	Kegel exercises	63	15.4
	Avoiding triggers	65	15.9
	Exercising	201	49.1
	Losing weight	139	34.0
	Reducing stress	96	23.5
	Herbal therapy	120	29.3
	Other lifestyle changes	0	0.0
Belief about lifestyle change effectiveness in perimenopause	Yes	161	39.4
	Maybe	132	32.3
	Some are	60	14.7
	Not sure	47	11.5
	No	9	2.2
Opinion about perimenopause education timing §	High school	185	45.2
	Medical routine checks	205	50.1
	University	189	46.2
	Other periods	18	4.4

§ A participant could provide more than one answer. ^⁂^ Frequencies of each symptom are depicted in Figure 3.

**Table 4 healthcare-12-00677-t004:** Demographic and health-related factors associated with knowledge score.

**Parameter**		**B**	**95% CI**	***p*-Value**
Age		0.07	−0.05; 0.19	0.223
Age at first menarche		−0.05	−0.37; 0.27	0.754
**Parameter**	**Level**	**Mean**	**SD**	***p*-Value**
Marital Status	Single	13.12	7.19	
	Married	14.94	5.47	
	Divorced	15.51	5.74	
	Widowed	13.20	4.07	0.151
Education	Illiterate/Primary	15.75	6.34	
	Middle school	13.31	4.87	
	Secondary school	13.33	5.83	
	University/Bachelor	15.87	5.01	
	Postgraduate	16.00	7.51	<0.001 *
Profession	Housewife	14.58	5.56	
	Employed	15.16	5.64	
	Retired or unemployed	14.71	6.82	0.602
Monthly income	No monthly income	14.44	5.98	
	<1 K SAR	13.28	4.86	
	1–5 K SAR	13.66	4.97	
	5–10 K SAR	16.37	5.48	
	10–15 K SAR	15.89	5.15	
	>15 K SAR	15.12	6.95	0.014 *
Parity	None	13.56	6.63	
	1–4	15.04	5.71	
	5 or more	14.94	4.87	0.207
Smoking status	Smoker	14.39	7.45	
	Ex-smoker	14.53	6.42	
	Passive smoker	16.32	5.94	
	Non-smoker	14.51	5.33	0.104
Contraceptive use	No	14.60	6.08	
	Yes	15.07	5.07	0.396
No. comorbidities	None	14.61	5.84	
	1	14.29	5.26	
	2+	16.40	5.50	0.031 *
Prescribed medication	No	14.70	5.77	
	Yes	14.95	5.50	0.658
Medication adherence	As prescribed	14.26	5.53	
	Sometimes skipping	15.90	5.12	
	Often skipping	18.22	5.78	
	Rarely as prescribed	14.33	7.23	
	Not applicable	14.70	5.77	0.149

B: linear regression coefficient; 95% CI: 95% confidence interval; otherwise one-way ANOVA was used. * Statistically significant difference (*p* < 0.05).

**Table 5 healthcare-12-00677-t005:** Association of knowledge about perimenopause with attitudes and practice.

Parameter	Level	Knowledge Score		*p*-Value
		Mean	SD	
Attitudes towards perimenopause	Excited	14.18	6.13	
	Resigned	14.70	6.01	
	Indifferent	14.34	5.21	
	Worried	15.71	5.85	
	Undecided	15.28	5.09	0.400
Visited doctor for perimenopause	Yes	16.27	5.66	
	No	14.29	5.55	0.002 *
Frequency of follow-up	Regularly	18.11	6.14	
	Often	17.24	4.86	
	Rarely	15.51	4.62	
	Never	12.76	5.68	<0.001 *
Symptoms experienced	None	14.69	5.98	
	1–2	12.37	4.92	
	3–5	13.83	4.64	
	>5	17.35	5.86	<0.001 *
No. of lifestyle measures practiced	None	11.52	5.26	
	1–2	13.84	5.49	
	3+	16.77	5.16	<0.001 *
Belief about lifestyle change effectiveness in perimenopause	Yes	15.45	5.65	
	Maybe	14.93	5.55	
	Some are	15.98	5.05	
	Not sure	11.40	5.34	
	No	11.89	5.40	<0.001 *

Test used: independent *t*-test or one-way ANOVA, as applicable. * Statistically significant difference (*p* < 0.05).

## Data Availability

The data used for the present study are available upon reasonable request from the corresponding author.

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
