# Peer review of "Knowledge, Awareness, Attitudes and Practices toward Perimenopausal Symptoms among Saudi Females"

_healthcare, 2024, doi:10.3390/healthcare12060677_

Round 1

Reviewer 1 Report

Comments and Suggestions for Authors

In this paper the authors aim at exploring the knowledge and attitudes in perimenopausal Saudi females through a cross-sectional survey conducted at the Ministry of Health (MOH) primary care centers (PHC) of Jeddah, Saudi Arabia.

Although the paper is overall well written there are some aspects in the statistical analysis that should be modified.

The authors use a questionnaire to evaluate awareness, knowledge and attitude; they report Cronbach's alpha for the knowledge dimension but no information on the overall construct validity (through an Exploratory Factor Analysis) of the questionnaire is provided. As this questionnaire was never validated before, I would recommend such analysis. When analyzing the association between demographic, health-related factors, attitudes and practice with knowledge score, the authors use several univariate analyses; however a more robust and informative approach would be the construction of a multivariable linearmodel in order to examine the independent association of each variable with knowledge score.   Minor remarks Please avoid expression like "highly significant association” as p-values cannot be considered as measure of the strength of association. Line 78: "between date 1 and date 2” Please, add the real dates .

Author Response

In this paper the authors aim at exploring the knowledge and attitudes in perimenopausal Saudi females through a cross-sectional survey conducted at the Ministry of Health (MOH) primary care centers (PHC) of Jeddah, Saudi Arabia. 

Although the paper is overall well written there are some aspects in the statistical analysis that should be modified.

Thanks for the appreciation and for the constructive comments and recommendations.

The authors use a questionnaire to evaluate awareness, knowledge and attitude; they report Cronbach's alpha for the knowledge dimension but no information on the overall construct validity (through an Exploratory Factor Analysis) of the questionnaire is provided. As this questionnaire was never validated before, I would recommend such analysis.

Thanks for this pertinent comment. Principal component analysis was carried out, in combination with Cronbach’s alpha, indicating acceptable unidimensionality. However, since the definitive construct validity of the knowledge questionnaire was not confirmed, the internal validity of the findings may be impacted. This was acknowledged in the limitation with a direction to further refine the questionnaire’s construct.

When analyzing the association between demographic, health-related factors, attitudes and practice with knowledge score, the authors use several univariate analyses; however, a more robust and informative approach would be the construction of a multivariable linear model in order to examine the independent association of each variable with knowledge score.  

Thanks for that advice. A Stepwise multivariate linear regression was conducted and showed interesting insights that were integrated in the Results, Discussion and Conclusions. Thanks a lot.

Minor remarks Please avoid expression like "highly significant association” as p-values cannot be considered as measure of the strength of association. Line 78: "between date 1 and date 2” Please, add the real dates.

We agree. The term was replaced by “statistically significant”. Thanks a lot.

Reviewer 2 Report

Comments and Suggestions for Authors

Dear Authors,

Thank you for the opportunity to read about your research. Perimenopause is a difficult time for many women. It's good that you decided to investigate women's knowledge and attitudes towards this period. I hope that the results, although they show that the level of knowledge of Sudi women is high, are a good reason to continue health education also among younger women in order to prepare them for changes in the functioning of the body and thus influence the quality of life.

I have a few doubts about organizing research:

Line: 78 – what means: between date 1 and date 2. When was the research conducted?

Line: 79-80 – information about the consent of the bioethics committee should be included in a separate paragraph

Simple size: I'm glad you explained how to calculate the sample size but you didn't provide the final sample size

Sampling: How were research respondents selected?

Data collection tool: how many questions were there in the questionnaire?

Procedure: please describe in more detail - how were the respondents selected, where did they fill out the form - in the corridor? How was anonymity ensured? Could they have dropped out during the study? Was a link to the online survey sent to an e-mail address or was there other access to the questions? How do you know who answered the online questionnaires?

  Table 1 – row – nationality is not necessary because the entire study concerns Saudi women

Author Response

Thank you for the opportunity to read about your research. Perimenopause is a difficult time for many women. It's good that you decided to investigate women's knowledge and attitudes towards this period. I hope that the results, although they show that the level of knowledge of Sudi women is high, are a good reason to continue health education also among younger women in order to prepare them for changes in the functioning of the body and thus influence the quality of life.

Thanks for your appreciation, encouragement and guidance for further health education.

I have a few doubts about organizing research:

Line: 78 – what means: between date 1 and date 2. When was the research conducted?

Indeed, this was an oversight; the study period was added in the revised version.

Line: 79-80 – information about the consent of the bioethics committee should be included in a separate paragraph

A subheading “Ethical Clearance” was added in the Methods section. Thanks.

Simple size: I'm glad you explained how to calculate the sample size but you didn't provide the final sample size

The target sample is specified to be 423 patients; however, the final sample size reached is specified in the Results section (409). Please advise whether further action is needed.

Sampling: How were research respondents selected?

Indeed, the sampling method (convenience) was not specified. It has been specified in the revised version.

Data collection tool: how many questions were there in the questionnaire?

The total number of questions was 46; this was added in the end of the Data collection tool subsection. Thanks

Procedure: please describe in more detail - how were the respondents selected, where did they fill out the form - in the corridor? How was anonymity ensured? Could they have dropped out during the study? Was a link to the online survey sent to an e-mail address or was there other access to the questions? How do you know who answered the online questionnaires?

Thanks for bringing these issues to light. All these issues were clarified in the revised version, under Procedure and Ethical Clearance subheadings of the Methods section. Thanks again.

  Table 1 – row – nationality is not necessary because the entire study concerns Saudi women

That is true; thanks for your relevant remark.Variable “Nationality” was removed. That bring the total number of questions to 45.

Round 2

Reviewer 1 Report

Comments and Suggestions for Authors

The authors have properly addressed my previous concerns and the paper is now suitable for publication.

Reviewer 2 Report

Comments and Suggestions for Authors

Dear Authors,

Thank you for using my tips.

Now your manuscript has an orderly, coherent structure, is legible and transparent.

Congratulations! Good job